# Beliefs, Practices and Health Care Seeking Behavior of Parents Regarding Fever in Children

**DOI:** 10.3390/medicina55070398

**Published:** 2019-07-22

**Authors:** Urzula Nora Urbane, Zane Likopa, Dace Gardovska, Jana Pavare

**Affiliations:** 1Department of Pediatrics, Riga Stradins University, Vienibas Gatve 45, LV-1004 Riga, Latvia; 2Department of Pediatrics, Children’s Clinical University Hospital, Vienibas Gatve 45, LV-1004 Riga, Latvia

**Keywords:** fever in children, parental beliefs, fever phobia, healthcare-seeking behavior, antipyretics

## Abstract

*Background and objectives:* Fever in children is one of the most common reasons for seeking medical attention. Parents often have misconceptions about the effects to fever, which leads to inappropriate use of medication and nonurgent visits to emergency departments (ED). The aim of this study was to clarify the beliefs on the effects and management of fever and to identify healthcare seeking patterns among parents of febrile children in Latvia. *Materials and Methods:* Parents and legal guardians of children attending ED with febrile illness were included in the study. Participants were recruited in Children’s Clinical University Hospital (CCUH) in Riga, and in six regional hospitals in Latvia. Data on beliefs about fever, administration of antipyretics, healthcare-seeking behavior, and experience in communication with health care workers were collected via questionnaire. *Results:* In total, 355 participants were enrolled: 199 in CCUH and 156 in regional hospitals; 59.2% of participants considered fever itself as indicative of serious illness and 92.8% believed it could raise the child’s body temperature up to a dangerous level. Antipyretics were usually administered at median temperature of 38.0 °C, and the median temperature believed to be dangerous was 39.7 °C; 56.7% of parents usually contacted a doctor within the first 24 h of the illness. Parents who believed that lower temperatures are dangerous to a child were more likely to contact a doctor earlier and out-of-hours; 60.1% of participants had contacted their family doctor prior their visit to ED. Parental evaluation of satisfaction with the information and reassurance provided by the doctors at the hospital was higher than of that provided by their family doctor; 68.2% of participants felt safer when their febrile children were treated at the hospital. *Conclusions:* Fever itself was regarded as indicative of serious illness and potentially dangerous to the child’s life. These misconceptions lead to inappropriate administration of antipyretics and early-seeking of medical attention, even out-of-hours. Hospital environment was viewed as safer and more reassuring when dealing with febrile illness in children. More emphasis must be placed on parental education on proper management of fever, especially in primary care

## 1. Introduction

In developed countries, up to 60% of children have experienced a febrile episode before the age of five years [1]. Fever is one of the most common reasons for seeking medical attention, constituting for up to 30% of visits to both primary care and pediatric emergency departments (ED) [2,3,4,5]. In up to 99% of cases fever is caused by self-limiting viral infections, and evidence suggests that fever itself is not dangerous and may even be beneficial for the immune response to infection. Elevated body temperature is the result of an increase in the hypothalamic set point, which is triggered by either microbial products or cytokines secreted by the host as a result of infection [6,7]. In a child with a healthy nervous system it very rarely rises above 42 °C, a body temperature that is associated with adverse effects [8,9,10]. On the contrary, studies suggest that fever aids the immune system during infection by promoting chemotaxis of neutrophils to the affected site, by inducing stress responses in microbes, and by intensifying the antiviral activity of Interferon (IFN) [11,12,13,14]. Therefore, as suggested by evidence-based guidelines, reduction of the child’s body temperature in case of fever is not always required, and should be reserved for situations when it causes distress [15].

As most febrile episodes in children are benign, assessment in primary care and recommendations given by a family doctor or general practitioner should largely be sufficient for management. However, fever remains one of the main reasons for nonurgent visits to out-of-hours healthcare and pediatric emergency departments [5,16,17,18,19,20]. It is also the most common reason for use of medication in children [15,21,22,23]. While evidence-based information is available on the appropriate measuring of fever, assessment of the child, and administration of antipyretics [15], studies on caregiver knowledge of fever show that the parental practices in management of fever in their children often deviate significantly from the guidelines [21,23,24,25,26]. In these studies, guardians have expressed many misconceptions on potential harmful effects of fever, resulting in the emergence of the term “fever phobia”. These misconceptions are most commonly the result of lack of knowledge on the pathophysiology and management of fever, although experience with serious illness in a child also makes some parents more alarmed about fever in their child. However, the prevalence of serious bacterial infections (SBI) in children with fever is low—from 1% in primary care to up to around 5–15% of children who present to ED—depending on the level of provided care and complexity of the conditions of referred patients [27,28].

Several qualitative studies show evidence that, in some occasions, increased parental anxiety in dealing with fever in a child is increased by ineffective communication with healthcare workers. In some cases, parents have felt that they are not taken seriously, and their complaints are not fully addressed but rather dismissed by stating that “it is nothing” [29]. In their opinion, clinicians often give incomplete explanation on the nature of the disease (“it’s just a virus”) without providing explanation on what it means, on how to evaluate the severity of the child’s condition, and when to seek help [30]. Parental misconceptions on hazards of fever are not sufficiently addressed in primary care, thus leading to visits to ED that could potentially be avoided [19,31].

Understanding parental concerns and beliefs is important for selecting the most appropriate educational measures. This study focuses on clarifying the beliefs on the effects and management of fever, and on identifying healthcare seeking patterns among parents of febrile children in Latvia. The influence of parental knowledge on fever and of their communication with clinicians on healthcare seeking behavior was also examined.

## 2. Materials and Methods

This was a cross-sectional study conducted in two parts. A convenience sample of parents or legal guardians who presented to the Emergency Department of Children’s Clinical University Hospital (CCUH) in Riga, the capital of Latvia, was recruited between October 2017 and December 2018. CCUH is the main pediatric hospital in the country providing tertiary level healthcare and around 8000 febrile children visit the ED each year, where they are assessed by pediatricians.

An additional sample of febrile patients and their parents presenting to ED was enrolled in six different regional hospitals in Latvia between January and March 2019. The hospitals included in the study were secondary care level hospitals providing 24 h emergency care in various specialties, including pediatrics. In these hospitals, there is a general ED admitting pediatric and adult patients alike, but care for children up to 18 years is provided by pediatricians.

All parents or legal guardians of patients aged 1 month up to 18 years presenting with fever were considered eligible, with exception of patients with severe chronic illnesses or immunosuppression. During their stay at the ED, the parents were approached with a questionnaire. Topics covered in the questions included beliefs about fever; administration of antipyretics; healthcare-seeking behavior, both when dealing with fever in their children in general and during the ongoing episode; and experience in communication with health care workers. Parental habits of seeking a doctor within normal working hours and out-of-hours were assessed. Normal working hours were defined according to the standards of National Health Service of Latvia as the time between 8 a.m. and 5 p.m. on working days, outside of which medical care is officially provided by out-of-hours primary care doctors or telephone service. The satisfaction of explanation and assurance after consulting a pediatrician was compared to that of other physicians consulted prior visiting the hospital (mostly the family doctor). Before implementation in this study, the questionnaire was piloted by a cohort of 26 patients.

In addition, demographic data (age and level of education of parents or legal guardians, number of children in the family, and age and gender of the patient admitted to ED) were also collected and analyzed. The contents of the questionnaire can be viewed in detail in Appendix A as Appendix A: Questionnaire of the view of parents/guardians on managing fever in children (English version).

Statistical analysis was performed using SPSS and MS Excel data analysis software. The results were summarized by applying descriptive statistics. The statistical significance of the differences between categorical variables was estimated by applying Pearson’s chi-squared test, odds ratios with 95% confidence intervals were calculated for comparison of variables between two groups using 2 × 2 contingency tables, and the Wilcoxon rank-sum test was used for comparison of two independent groups of nonparametric data. A significance level of *p* < 0.05 was applied.

Informed consent for enrolment in the study and analysis of the data was obtained for each participant. The study was approved by the Ethics Committee of Riga Stradins University, approval No. 13/05.10.2017 for enrollment of CCUH cohort and No. 6-3/27 22.10.2018 for enrollment in regional hospitals.

## 3. Results

### 3.1. Demographic Data

In total, 355 respondents were enrolled in the study, of them 199 were enrolled in CCUH and 156 were recruited in the regional hospitals. The complete dataset containing answers of the respondents to the questionnaire can be viewed in Appendix A.

#### 3.1.1. Participants in Children’s Clinical University Hospital

In CCUH 88.5% of participants who completed the questionnaire (*n* = 176) were mothers aged 21–56 years (median age 34 years), 9.0% (*n* = 18) were fathers aged between 23 and 52 years (median age 34 years), and 2.5% (*n* = 5) were other guardians (mostly grandparents). The level of education of participants in CCUH was higher than in regional hospitals (OR (95% CI) = 1.7(1.1–2.6), *p* = 0.019), with 49% of them having higher education (bachelor’s degree or higher).

The children of participants who were admitted to ED during the time of study were aged 3 months to 17 years and 10 months (median age 48 months). 51.8% of the patients were boys (*n* = 103). 31.2% (*n* = 62) patients developed serious bacterial infections (SBI) (defined for this study as bacterial meningitis, sepsis, bacteremia, pneumonia (positive consolidation on chest X-ray), urinary tract infection (positive urine culture and microscopy), bacterial gastroenteritis (positive bacterial pathogen in stool), appendicitis, and osteomyelitis); 61.8% (*n* = 123) were hospitalized and 58.8% (*n* = 117) received antibacterial treatment.

#### 3.1.2. Participants in the Regional Hospitals

In regional hospitals altogether, 156 respondents completed the questionnaire, 93.0% (*n* = 146) of whom were mothers aged 18–48 years (median age 31 years), 5.7% were fathers (*n* = 9) aged 30–43 years (median 34 years), and 2 were grandparents; 36.4% of respondents had higher education. Most of the families enrolled in both CCUH and regional hospitals had one or two children, but the families having three or more children were significantly more common among the participants in the regional hospitals (OR (95% CI) = 2.1(1.2–3.9), *p* = 0.009). The number of children in the families of participants is reflected in Figure 1.

The febrile patients of the participants visiting the ED for the ongoing febrile episode were one month to 16 years and 4 months old; the median age was 27 months. The percentage of boys was 47.4% (*n* = 74); 24.4% (*n* = 38) of these patients were diagnosed with SBI, 89.7% (*n* = 139) were hospitalized, and 61.2% (*n* = 61.3%) received antibacterial treatment.

The birth order of the children of participants admitted to ED during the study is displayed in Figure 2.

### 3.2. Beliefs Regarding Fever

The majority (59.2%) of participants considered fever itself as indicative of serious illness; 27.9% of parents stated that other symptoms should be considered as well to evaluate the illness as serious. Only 9.4% of participants thought that fever alone is not indicative of severity of illness, while 4.3% did not know whether or not it was so. The differences between the opinions stated by participants enrolled in CCUH and regional hospitals can be viewed in Figure 3. There were no significant differences between the beliefs of parents with different education levels or families with one or multiple children.

The body temperature of the child at which parents usually administered antipyretics ranged from 37.0–40 °C, with a median of 38 °C. Half of the respondents (50.1%) gave antipyretics at 38 °C, while one-third of patients (33.1%) gave medication at 38.5 °C. Only 7.6% of participants reported allowing the temperature to rise above 39 °C, while 9.2% stated that they start reducing the child’s body temperature before it reaches 37.9 °C. Respondents with higher education (bachelor’s degree or higher) gave medication to reduce fever at a higher temperature (median 38.5 °C) than parents without higher education (median 38 °C), the difference was statistically significant (*p* < 0.001). The number of children in the family (one or multiple) did not significantly affect the temperature at which antipyretics were given, and the practices between respondents in CCUH and regional hospitals were similar.

The median temperature that parents evaluated as high fever in CCUH and regional hospitals alike was 39 °C. The vast majority of respondents (92.8%) believed that the child’s body temperature during febrile illness can increase up to a level that is dangerous to the child’s life. The median temperature believed to be dangerous to the child by all respondents was 39.7 °C, though there were differences between the study sites. While among respondents in CCUH, median temperature associated with adverse effects was 39.5 °C, parents in regional hospitals mostly regarded fever above 40 °C as threatening, though the difference was not statistically significant (*p* = 0.11). Neither level of education nor family size affected parental beliefs on temperatures regarded as high fever or dangerous to the child (*p* > 0.05).

### 3.3. Healthcare Seeking Behavior

Slightly more than a half of the participants (56.7%) admitted that they seek medical attention within the first 24 h after their children become ill with fever (54.4% of participants in CCUH and 59.7% of respondents in regional hospitals). The time after the onset of febrile illness when parents usually sought help is reflected in Figure 4.

Parents of a single child were slightly more likely to seek medical attention within the first 24 h than parents of multiple children (OR (95%CI) = 1.65 (1.05–2.61), *p* = 0.03, χ^2^ = 4.67). The median temperature believed to dangerous was lower for parents seeking help within the first 24 h (median 39.5 °C) than for parents who usually seek help later (median 40 °C), though the difference was not statistically significant (*p* = 0.09). Similarly, parents who usually sought help on the first day of illness were also giving their children antipyretics at a lower body temperature (median 38 °C) than parents who waited until later (median 38.3 °C) (*p* = 0.009). The education level of respondents did not affect the time at which they usually contacted a doctor when coping with febrile illness in their child.

When asked when they first contacted a doctor during the current febrile episode, 51.4% of participants did so within the first 24 h after the onset of symptoms, and the number of children in the family did not correlate with seeking help earlier or later, neither did the education level of the parents. The body temperature associated with adverse effects was lower (median 39.5 °C) among parents who sought help on the first day than among those who did so later (median 40 °C) (*p* = 0.003).

The first doctor visited or contacted during the ongoing febrile episode by majority of participants in CCUH and regional hospitals (68.6%) was a primary care specialist (in 60.1% of cases it was the family doctor, 7.4% contacted the out-of-hours family doctor telephone service, while 1.1% of participants visited an out-of-hours primary care doctor). Participants enrolled in CCUH more commonly were first seen by an ambulance doctor or physician at the hospital (31.7%) than respondents in regional hospitals, whose children in only 23.7% of cases were first examined by these specialists. More details can be viewed in Figure 5.

The first attempt of seeking medical attention within the ongoing episode was mostly (in 62.2% of cases) within the normal working hours (8 a.m. to 5 p.m. on working days). In regional hospitals, 42.1% of participants first visited or called a doctor outside the normal working ours, compared to 34.4% of parents enrolled in CCUH.

36.6% of patients who first called the ambulance or went to the hospital did so within the normal working hours, with a marked prevalence among the CCUH cohort (45.8%) over parents filling the questionnaire in regional hospitals, where only 20.6% first sought help outside primary care within the working hours. 

The median temperature believed to be dangerous by participants who sought help outside the working hours was lower (median 39.5 °C) than that believed to be harmful by those who first visited or called the doctor within the working hours (median 39.9 °C), though this difference was not statistically significant (*p* = 0.07).

### 3.4. Satisfaction with Provided Care

The satisfaction with the explanation on the nature of illness provided by the doctor at the emergency department of the hospital was higher than with that given by the family doctor. This was true among the CCUH and regional hospital cohorts alike. Respondents in regional hospitals were more satisfied with the information provided by both the family doctors and the doctors at the emergency department of the hospital when compared with parents whose children visited CCUH (Figure 6).

Similarly, parental concern was reduced more effectively after a consultation with the physician at the emergency department than after the visit or call to the family doctor, and participants at the regional hospitals evaluated the effect of both consultations on their level of concern more positively than the CCUH cohort (Figure 7); 68.2% of all participants stated that when dealing with febrile illness in their child they feel safer than under the care of their family doctor, while 29.5% were unsure, and 2.3% felt safer when treated by the family doctor.

The majority of participants (66.1%) evaluated the availability of their family doctor as “good” or “very good”. The satisfaction was significantly higher among participants in regional hospitals, where 72% assessed the availability as “good” or “very good”, when compared to respondents in CCUH, where 59% rated it so (*p* = 0.004). Of those who sought medical assessment within the working hours, this evaluation was given by 62.6%, while among those who visited or called a doctor outside normal working hours it was 71.1%. There was no statistically significant difference in satisfaction with the availability of family doctor between respondents who first contacted a primary care specialist and those whose children were assessed for the first time by the clinicians at the ambulance or at the emergency department of the hospital.

## 4. Discussion

The results of the questionnaire showed evidence of fever phobia among participants. Most parents believed that fever itself is indicative of serious illness, and that the child’s body temperature can increase to a level that could possibly endanger the child’s life. The belief that fever itself is a disease rather than a symptom has been prevalent for decades [21,24], leading to many misconceptions and sense of urgency in parents to reduce it [26,32].

Naturally, attempts to reduce the child’s body temperature as soon as it reached 38 °C were common among the study participants, though parents with higher education more commonly delayed giving antipyretics until it was higher than 38.5 °C. Some parents (9.2%) would even give antipyretics before the temperature reached 37.9 °C. These practices contradict the advice given in several evidence-based guidelines [15,33,34,35], which state that antipyretic are not always necessary in case of fever, and should be reserved for cases when the child is feeling significant discomfort. However, the low threshold of giving medication to reduce fever is not unique to parents in Latvia, as other studies in the United States, Israel, Australia, and Italy [21,26,31,36,37,38], where the proportion of parents giving antipyretics before the temperature reached 38 °C ranged from 2% to more than a half. The median temperature believed to pose a threat to the child’s life among our study participants was similar to that stated by parents in other countries [25,26], though there were reports of temperatures as low as 38–39 °C presumed to cause serious health risks [39].

Beliefs on fever affected the healthcare seeking behavior of the study participants—parents who believed that lower temperatures are dangerous to a child were more likely to contact a doctor earlier, even outside normal working hours. Also, parents who usually sought help within the first 24 hours of the onset of febrile illness were used to giving antipyretics at a lower body temperature than those who believed that a consultation by a healthcare specialist could be delayed until later. Fever is one of the main reasons for seeking healthcare specialists after hours [19,21], despite the fact that many of these consultations are nonurgent and should be managed in primary care.

It was also evident that the respondents of the questionnaire were more satisfied with explanatory work by doctors at the hospital than what they previously received at their family doctors. The majority also felt safer in the hospital than under the care of their family doctors. Almost two thirds of the participants had contacted their family doctor during the ongoing episode of child’s illness, but sought help at the ED nevertheless. Though the study only included patients who eventually visited the ED of a hospital and did not assess the opinion of patients who were only treated in primary care, this shows that incomplete success of reducing parental concern on a febrile illness in their child in primary care may lead to them seeking help elsewhere. Similarly, another study conducted in Tel Aviv revealed that many parents still had misconceptions about fever despite visiting the general practitioner within two days before seeking help at the ED, and the anxiety caused by fever in their child was not lower than in parents who had not been consulted in primary care prior the visit to ED [31].

The satisfaction levels of parents in CCUH and regional hospitals with provided information and reassurance were not 100% after visiting either the family doctor or the specialist working at the ED, which indicates that communication with parents, including education on the nature and management of febrile illness, needs improvement in both primary care and hospitals.

There were marked differences between the study cohorts regarding seeking medical attention. The participants recruited in regional hospitals were less likely to skip primary care within normal working hours than the CCUH cohort; their satisfaction with the availability of family doctor was higher, as was their contentment with provided information and the ability of the family doctor to reduce their anxiety. Parents in regional hospitals also attributed adverse effects to higher temperatures than the CCUH, though the habits of administering antipyretics were similar. Whereas among the parents enrolled CCUH, the evaluation of the availability of the family doctor was not as high as among parents in regional hospitals, and more parents turned to ambulance or emergency department without consulting primary care first. Of those who contacted the family doctor before going to hospital, the evaluation of the communication with the physician was lower than in the other cohort. The reasons behind this were not investigated via the short questionnaire, however it can be concluded that, with the aim of increasing parental confidence and reducing the number of patients visiting the ED for febrile illness, more emphasis must be placed on improving the quality of support provided in primary care.

Several educational strategies have been proposed and studied in other countries, including face-to-face educational sessions, written handout materials with illustrations, educational videos and websites, and many have shown promising results in improving parental knowledge and understanding of febrile illness in children [29,40,41]. Similar measures must be taken for educating the parents in Latvia; this could be achieved by creation of a universal guidance including evidence-based and easily understandable information, which could be distributed by doctors in primary care and emergency departments, as well as made available online.

This study has some limitations, which we are aware of. First, only patients visiting the emergency department were enrolled, thus limiting the applicability of the results on the general population, in which many febrile children are successfully treated in primary care. Hospital settings were selected with the aim of recruiting patients originating from the capital and various other regions in Latvia, and to get insight in the reasons why parents choose to visit the ED in case of febrile illness in their children. Secondly, in order to limit the length of the questionnaire, specific details on the factors associated with parental anxiety in case of fever in children, as well as on parental experience in communication with healthcare workers, were omitted. To clarify this information, a qualitative interview study on parental coping with febrile illness was conducted [42]. Finally, no standardized method of measuring body temperature was applied to this study. As this study aimed to evaluate general beliefs of parents, the research team did not impose a change in measurement practices the participants commonly used at home, in which there were different variations. Nevertheless, the applied study design provides an adequate amount of information to address the research question and aim of the study.

## 5. Conclusions

Parents in Latvia often regard fever itself as indicative of serious illness and believe it could potentially endanger the child’s life. These misconceptions lead to management practices that contradict evidence-based guidelines and early-seeking of medical attention, even out-of-hours. Parents view hospital environment and specialist care as safer and providing more reassurance than primary care when coping with febrile illness. More emphasis must be placed on parental education on proper management of fever, especially in primary care.

## Figures and Tables

**Figure 1 medicina-55-00398-f001:**
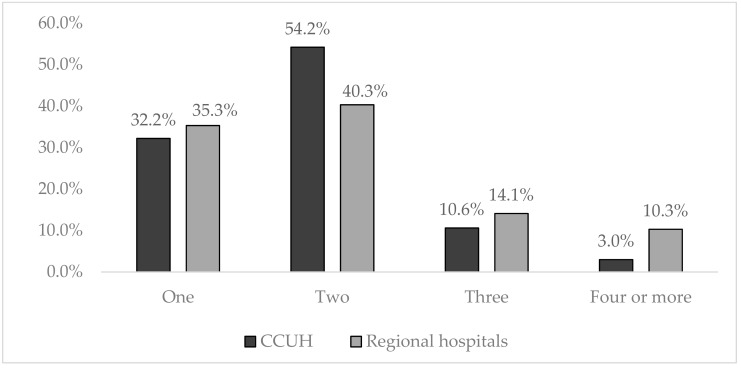
Number of children in the families of the participants. The proportion of families having three or more children was significantly higher among the participants in the regional hospitals (OR (95% CI) = 2.1(1.2–3.9), *p* = 0.009).

**Figure 2 medicina-55-00398-f002:**
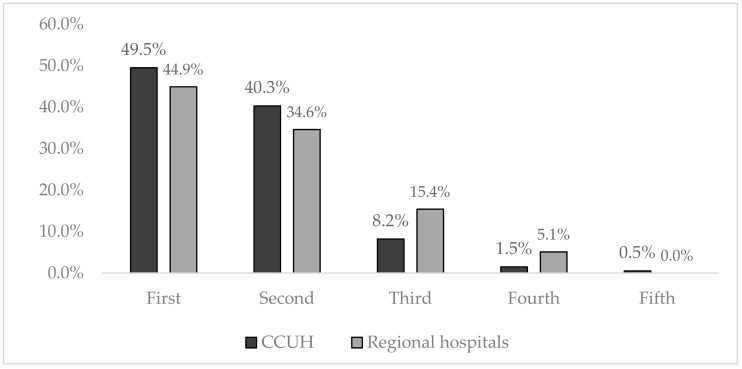
Birth order of the children of the participants admitted to emergency departments (ED) during the study.

**Figure 3 medicina-55-00398-f003:**
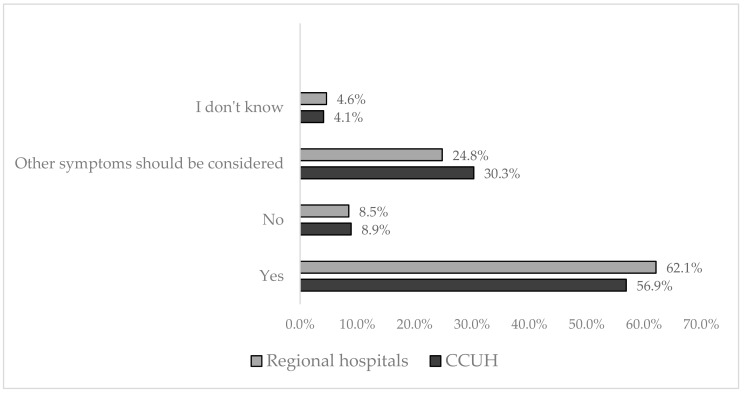
Parental response to question: “Does fever itself indicate that the illness is serious?”.

**Figure 4 medicina-55-00398-f004:**
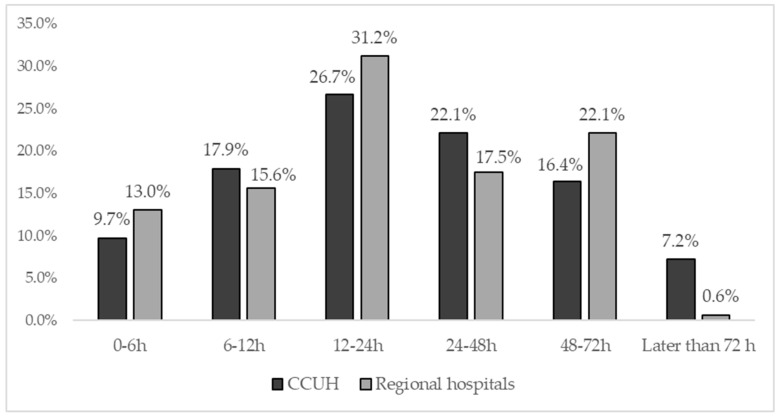
Time after the onset of febrile illness at which parents usually seek medical attention.

**Figure 5 medicina-55-00398-f005:**
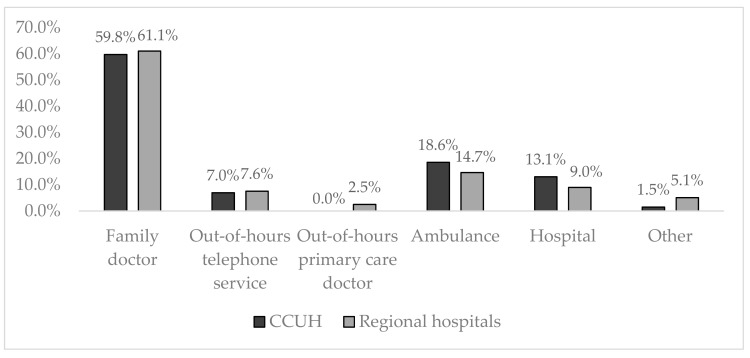
First doctor visited or contacted after the onset of symptoms of the ongoing febrile episode.

**Figure 6 medicina-55-00398-f006:**
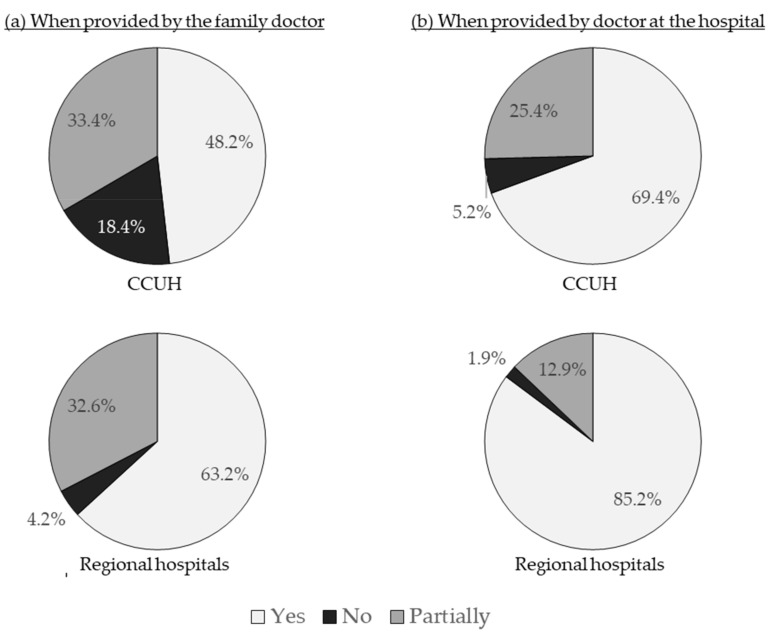
Was the explanation on the nature of illness and reasons for fever satisfactory? (**a**) Evaluation of the explanation provided by family doctors (applicable to 59.8% of participants in CCUH and 61.1% of participants in regional hospitals); (**b**) Evaluation of explanation provided by the pediatrician at the hospital (applicable to all participants).

**Figure 7 medicina-55-00398-f007:**
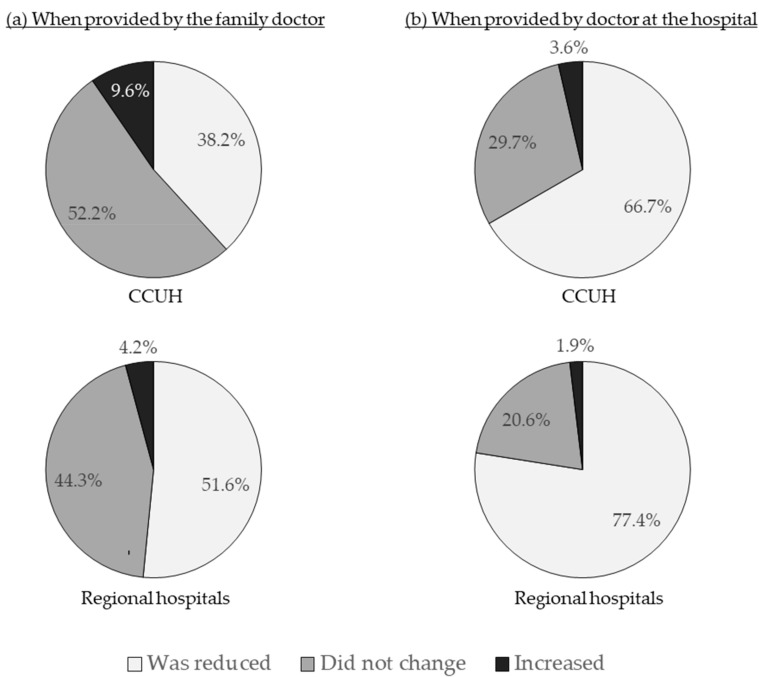
How did the information provided by the doctor affect your level of concern about the illness of your child? (**a**) Evaluation of assurance provided by family doctors (applicable to 59.8% of participants in CCUH and 61.1% of participants in regional hospitals); (**b**) Evaluation of assurance provided by the pediatrician at the hospital (applicable to all participants).

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
