# Peer review of "Beliefs, Practices and Health Care Seeking Behavior of Parents Regarding Fever in Children"

_medicina, 2019, doi:10.3390/medicina55070398_

Round 1
Reviewer 1 Report
Please be so kind to make a comment regarding:
- the measurement for fever.
- There are different type of measuring devices? Possible error in reported values?
Author Response
Response to reviewer 1 comments
Point 1. Please be so kind to make a comment regarding:
- the measurement for fever.
- There are different type of measuring devices? Possible error in reported values?
Response. Thank you for your comment. The aim of this study was to evaluate parental beliefs regarding fever, in which the research team had to rely on the past experiences of the participants. There were variations in the methods used by parents to measure their child’s body temperature, and before the questionnaire it was not possible to introduce a universal method to be used by all participants. We are aware that, as body temperature readings are affected by the site and method of measurements, this may affect the interpretation of body temperatures regarded as dangerous or requiring medication. Therefore, the following sentence has been added to limitations in the discussion section of the manuscript (lines 330 - 333 on page 11):
“Finally, no standardized method of measuring body temperature was applied to this study. As this study aimed to evaluate general beliefs of parents, the research team did not impose a change in measurement practices the participants commonly used at home, in which there were different variations.”
Reviewer 2 Report
The authors present a prospective study to determine the beliefs on the effects and management of fever, and to identify healthcare seeking patterns among parents of febrile children in Latvia. The study is interesting with a direct clinical application.
MAJOR COMMENTS
1. It is not clear if the specialized care at the hospitals is provided by family doctors or by pediatricians. It is necessary to explain better the medical care organization of the emergency departments in Latvia Hospitals. Do they have Pediatric ED or General ED? What are the differences between CCUH and regional hospitals in ED organization? Are there pediatricians in the ED of the hospitals? If so, is there any difference between the pediatricians, general doctors and the family doctors in terms of parent’s satisfaction?
2. Description of the sample: It could be interesting to describe the clinical characteristics of the children sample in terms of final diagnosis and evolution. How many children were admitted to the Hospital?, How many did have serious diseases?
3. Discussion: the authors say that “Several educational strategies have been proposed and studied in other countries” ant their proposal is to incorporate them in Latvia. Is there currently any educational programme in Latvia to improve parental knowledge of febrile illness in children?
MINOR COMMENTS
1. Definition of out-of-hours (8am to 5pm) is necessary in Methods section.
2. Explanation of the healthcare levels in Latvia. “CCUH provides tertiary level healthcare (level IV). Is it the same tertiary care level and level IV?
3. Are there pediatricians in the ED of regional hospitals?
4. Figures. It is necessary to explain in the caption of the figures if there is any significant difference in the variables that were shown in the graphics.
5. Results. Beliefs regarding fever: if the difference was not statistically significant the authors cannot say that “parents enrolled in CCUH were more likely to consider other clinical symptoms when assessing the seriousness of the illness and less likely to think that fever alone means it is severe that participants from the regional hospitals”. In general, I think that if the difference is not significant there is no difference, except if you consider that the sample size could not permit to reach possible statistical differences.
6. Page 5. Line 156. It is not necessary to explain “the difference was statistically significant” and include the value w=18295, It is enough to include “p<0.001”. The same in other paragraphs of the manuscript.
Author Response
Response to reviewer 2 comments
MAJOR COMMENTS
Point 1. It is not clear if the specialized care at the hospitals is provided by family doctors or by pediatricians. It is necessary to explain better the medical care organization of the emergency departments in Latvia Hospitals. Do they have Pediatric ED or General ED? What are the differences between CCUH and regional hospitals in ED organization? Are there pediatricians in the ED of the hospitals? If so, is there any difference between the pediatricians, general doctors and the family doctors in terms of parent’s satisfaction?
Response. Thank you for the comment. Medical care of children up to 18 years at the hospitals in Latvia is provided by pediatricians. In CCUH, the only specialized pediatric hospital of Latvia, there is a pediatric ED providing care to children only. In regional hospitals, there is a general ED admitting children and adults alike, however all children are assessed by pediatricians. No specialist care at the hospitals is provided by family doctors, who work in primary care. To make this clear in the main manuscript text, the text in Methods section has been altered to include the following information (lines 82 to 88 on page 2, and lines 98 to 100 on page 3):
“CCUH is the main pediatric hospital in the country providing tertiary level healthcare, and around 8000 febrile children visit the ED each year, where they are assessed by pediatricians.
An additional sample of febrile patients and their parents presenting to ED was enrolled in six different regional hospitals in Latvia, between January and March 2019. The hospitals included in the study were secondary level hospitals providing 24-hour emergency care in various specialties, including pediatrics. In these hospitals, there is a general ED admitting pediatric and adult patients alike, but care for children up to 18 years is provided by pediatricians.
[..] The satisfaction of explanation and assurance after consulting a pediatrician was compared to that of other physicians consulted prior visiting the hospital (mostly the family doctor).”
Point 2. Description of the sample: It could be interesting to describe the clinical characteristics of the children sample in terms of final diagnosis and evolution. How many children were admitted to the Hospital?, How many did have serious diseases?
Response. Thank you for the suggestion. The requested information has been added to the manuscript for CCUH (page 3, lines 130 - 134):
“31.2% (n=62) patients developed serious bacterial infections (SBI) (defined for this study as bacterial meningitis, sepsis, bacteremia, pneumonia (positive consolidation on chest X-ray), urinary tract infection (positive urine culture and microscopy), bacterial gastroenteritis (positive bacterial pathogen in stool), appendicitis, and osteomyelitis). 61.8% (n=123) were hospitalized, and 58.8% (n=117) received antibacterial treatment.”
The following information has been added to describe patients in regional hospitals (page 4, lines 145 - 146):
“24.4% (n=38) of these patients were diagnosed with SBI, 89.7% (n=139) were hospitalized, and 61.2% (n=61.3%) received antibacterial treatment.”
The authors note that Supplementary file 2 has been updated to contain these data.
Point 3. Discussion: the authors say that “Several educational strategies have been proposed and studied in other countries” ant their proposal is to incorporate them in Latvia. Is there currently any educational programme in Latvia to improve parental knowledge of febrile illness in children?
Response. Thank you for your question. There have not been any official educational programmes in Latvia to improve parental knowledge and understanding of fever so far. However, new clinical pathway and algorithm for management of febrile illness in children has been developed and approved by the Ministry of Health of Latvia a week ago, which contains educational sections for parents (and also clinicians) on evaluation of a child with fever, appropriate administration of antipyretics, and recognizing and responding to fever phobia. The group of authors have been closely involved in creation of this material, and the results of this study have contributed to adjustments for meeting parental informational needs.
MINOR COMMENTS
Point 1. Definition of out-of-hours (8am to 5pm) is necessary in Methods section.
Response: Thank you for the suggestion. The definition has been added to the revised version of the manuscript in lines 94 – 98 on pages 2 and 3, which now read as follows:
“Parental habits of seeking a doctor within normal working hours and out-of-hours were assessed. Normal working hours were defined according to the standards of National Health Service of Latvia as the time between 8 am and 5 pm on working days, outside of which medical care is officially provided by out-of-hours primary care doctors or telephone service.”
Point 2. Explanation of the healthcare levels in Latvia. “CCUH provides tertiary level healthcare (level IV). Is it the same tertiary care level and level IV?
Response: Thank you for the remark. We are now aware that this description causes confusion in the readers.
To clarify the levels of provided care of Latvian hospitals, it has been classified in the following way according to the Ministry of Health:
Level I: hospitals providing care for non-life-threatening emergencies in internal medicine, mild trauma, and chronic illnesses.
Level II: providing care in the following departments: internal medicine, chronic illnesses, surgery, neurology, obstetrics and gynecology, and pediatrics, as well as emergency medicine.
Level III: In addition to services described in level 2, these hospitals provide specialist care in specific departments (such as invasive cardiology, oncology, etc.)
Level IV: University hospitals providing tertiary healthcare, may specialize in specific departments.
As this classification is local and subject to change in the following years, and to make the differences between CCUH and regional hospitals more understandable, we have appropriately described CCUH as a tertiary hospital, and regional hospitals as providing secondary healthcare (lines 82 - 88 on page 2):
“CCUH is the main pediatric hospital in the country providing tertiary level healthcare, and around 8000 febrile children visit the ED each year, where they are assessed by pediatricians.
An additional sample of febrile patients and their parents presenting to ED was enrolled in six different regional hospitals in Latvia, between January and March 2019. The hospitals included in the study were secondary care level hospitals providing 24-hour emergency care in various specialties, including pediatrics. In these hospitals, there is a general ED admitting pediatric and adult patients alike, but care for children up to 17 years is provided by pediatricians.”
Point 3. Are there pediatricians in the ED of regional hospitals?
Response: Thank you for the question. Yes, there are, as stated above.
Point 4. Figures. It is necessary to explain in the caption of the figures if there is any significant difference in the variables that were shown in the graphics.
Response. Thank you for the remark. A caption explaining the differences has been added after Figure 1: “The proportion of families having three or more children was significantly higher among the participants in the regional hospitals (OR (95% CI) = 2.1(1.2-3.9), p=0.009).”
However, the group of authors is hesitant to place captions after each figure, as most of them are more illustrative in nature, or the explanation is not derived from the data displayed in the figure, such as the reflection of data analysis following Figure 4.
Point 5. Results. Beliefs regarding fever: if the difference was not statistically significant the authors cannot say that “parents enrolled in CCUH were more likely to consider other clinical symptoms when assessing the seriousness of the illness and less likely to think that fever alone means it is severe that participants from the regional hospitals”. In general, I think that if the difference is not significant there is no difference, except if you consider that the sample size could not permit to reach possible statistical differences.
Response. Thank you for the remark. The sentence has been removed from the revised version of the manuscript.
Point 6. Page 5. Line 156. It is not necessary to explain “the difference was statistically significant” and include the value w=18295, It is enough to include “p<0.001”. The same in other paragraphs of the manuscript.
Response. Thank you for the remark. This has been corrected wherever necessary.